# From Scaling Law to Sub-Scaling Law: Understanding the Diminishing Returns of Larger Models

## Abstract

Traditional scaling laws suggest that performance metrics of language models improve predictably with increases in model or dataset size. However, recent works display sub-scaling growth for large language models, where performance improvements decelerate as the dataset or model size increases. This study aims to systematically investigate the sub-scaling law phenomenon through an extensive empirical analysis involving over 400 models, ranging from 20 million to 7 billion parameters, with varying datasets and training strategies. Our findings indicate that sub-scaling laws arise primarily from high data density and non-optimal training resource allocations. Specifically, we observed that both factors contribute more significantly to performance deceleration than previously anticipated. We examine the sub-scaling phenomenon from two perspectives: data density and training strategy. High data density leads to diminishing marginal gains in performance, while optimal resource allocation is crucial for sustaining performance improvements. Further, we propose a sub-optimal scaling law that generalizes the Chinchilla scaling law to better predict performance and loss in sub-scaling regimes.

## 1 Introduction

The rapid advancement in natural language processing (NLP) has been significantly driven by the development of increasingly large language models. These models, such as LLaMA (Touvron et al., 2023), Chinchilla (70B) (Hoffmann et al., 2022), Gopher (280B) (Rae et al., 2021), and Megatron-Turing NLG (530B) (Smith et al., 2022), have set new benchmarks across a variety of linguistic tasks. There is also a growing body of research on scaling strategies (McCandlish et al., 2018; Yang et al., 2022; 2023), which could be beneficial for large language models (LLMs). The conventional wisdom suggests that augmenting model size and corresponding training data generally results in enhanced performance. This trend has led to the popularization of a 'bigger is better' paradigm within the field. This scaling up has been driven by the empirical observation that larger models trained on vast amounts of data tend to perform better on various natural language processing tasks (Brown et al., 2020; Komatsuzaki, 2019; Hernandez et al., 2022a).

However, recent empirical studies Hernandez et al. (2022a); Hu et al. (2023); Porian et al. (2024); Muennighoff et al. (2024) have observed deviations from this expected trend, particularly in the context of exceptionally large language models. These deviations manifest as sub-scaling growth, where the rate of performance improvement decelerates as model or dataset size continues to increase. Specifically, Hernandez et al. (2022a); Muennighoff et al. (2024) observe that sub-scaling occurs in scenarios involving repeated training data, leading to diminishing returns in performance. Hu et al. (2023) highlight that sub-scaling is particularly pronounced in tasks requiring complex reasoning or multi-step processes. Furthermore, Porian et al. (2024) find that sub-scaling exists under non-optimal training strategies with sub-optimal hyper-parameters. Figure 1

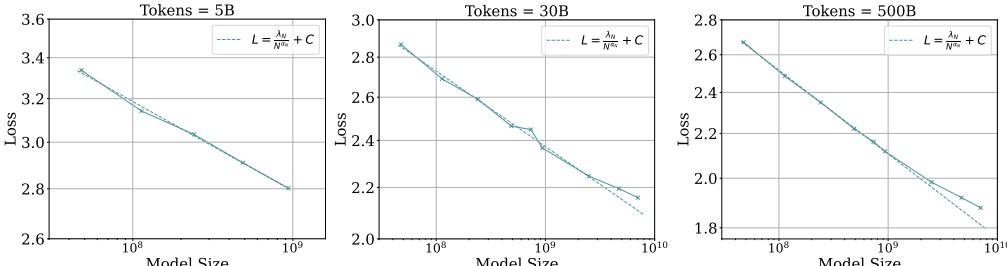

Figure 1: Sub-scaling phenomenon in loss. Scaling law fits well with 5B training tokens, but as tokens increase, loss curve shows greater curvature, and fitting accuracy decreases, especially for larger models.

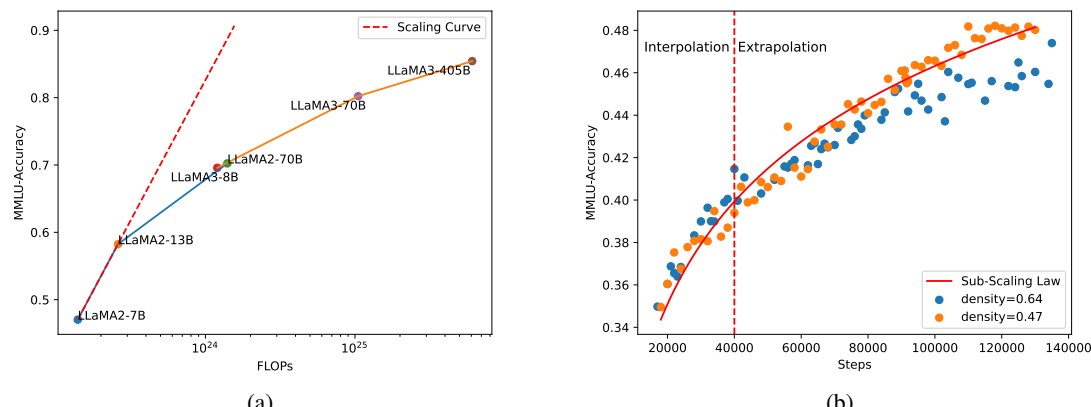

Figure 2: (a) LLaMA 2's scaling curve outperforms LLaMA 3's, despite LLaMA 3's advanced strategies. (b) Higher density datasets lead to sub-scaling. We propose metric density to measure redundancy and diversity: higher density indicates more redundancy and less diversity, leading to sub-scaling (see Section 2.1).

provides a visualization of the diminishing returns, clearly showing that as training progresses, the actual training loss values tend to be higher than those extrapolated from earlier stages, indicating how traditional scaling laws fall short when dealing with extensive datasets and suggests the need for a modified approach. Moreover, Hernandez et al. (2022a); Muennighoff et al. (2024) have similar sub-scaling observations in repeated data and non-optimal training strategy. However, there is a lack of systematic research on the sub-scaling behavior of large language models (LLMs).

Further extending this observation to model performance, Figure 2 displays the results of our tests on the performance scaling law Yang et al. (2024); Isik et al. (2024); Wu & Tang (2024) with LLaMA 2 and LLaMA 3 models. Despite LLaMA 3 incorporating advanced training strategies and improved data quality, the performance improvements from LLaMA 2 to LLaMA 3 decelerate as the training flops increase, LLaMA 2 with 70B parameters outperforms LLaMA 3 with 8B parameters.This discrepancy, depicted in Figure 2(a), underscores the inadequacies of traditional scaling laws. Additionally, when the scale of training data surpasses an optimal threshold relative to the available computational resources, sub-scaling law happens with such over-training (Gadre et al., 2024), potentially leading to diminishing returns in model performance. Moreover, there is a lack of understanding of the training dynamics of large language models and the sub-scaling laws governing the training strategies of language models. This motivates the question: *Under what conditions do sub-scaling laws influence the performance and efficiency of large language models?*

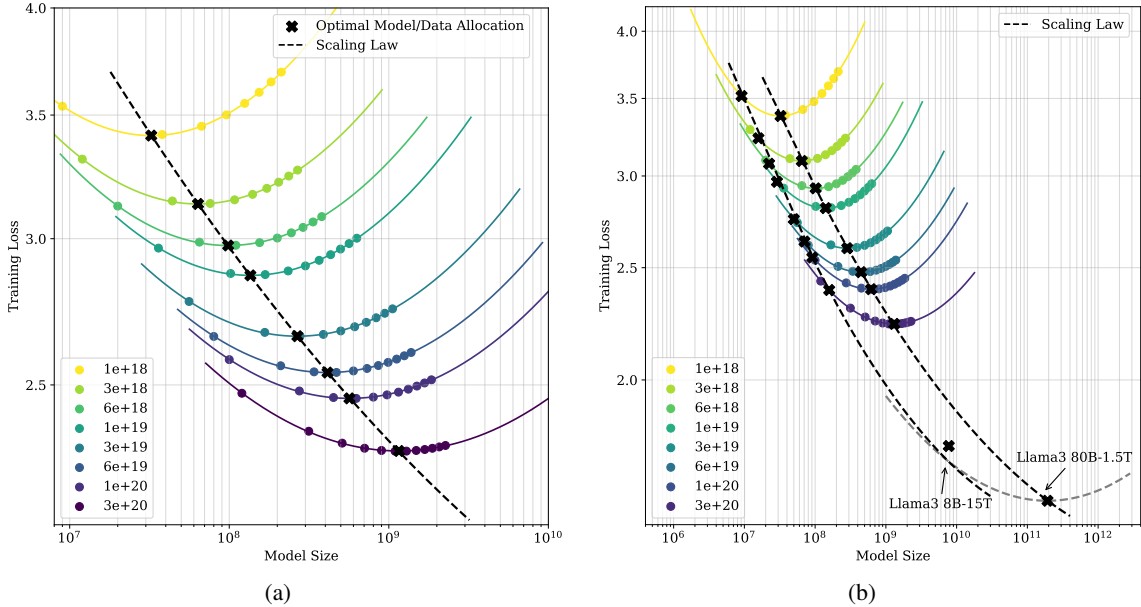

Figure 3: (a) With a fixed total compute budget, we adjust the model-to-data allocation ratio and plot the training loss against model size. A black curve connects the minimum points of each curve, illustrating the optimal Chinchilla law. (b) However, current large language models, such as Llama3 8B, are trained on 15T tokens, with a model-to-data allocation strategy that significantly deviates from the optimal Chinchilla law.

This study aims to systematically investigate the sub-scaling law phenomenon through an extensive empirical analysis involving over 400 models, ranging from 20 million to 7 billion parameters, with varying datasets and training strategies. Our findings indicate that sub-scaling laws arise primarily from high data density and non-optimal training resource allocations. Specifically, we observed that both factors contribute more significantly to performance deceleration than previously anticipated. We examine the sub-scaling phenomenon from two perspectives: data density and training strategy. High data density leads to diminishing marginal gains in performance as shown in Figure 2, while optimal resource allocation is crucial for sustaining performance improvements as shown in Figure 3. Further, we propose a sub-optimal scaling law that generalizes the Chinchilla scaling law Hoffmann et al. (2022) to better predict performance and loss in sub-scaling regimes. Our analysis reveals that the quality and diversity of training data are paramount, often outweighing the benefits of mere scale in model size. Key findings from our study include:

1. *Sub-Scaling Law Phenomenon*: Traditional scaling laws fail to predict performance improvements for very large models and datasets. The performance gains decelerate, leading to sub-scaling growth, especially in high data density scenarios and with non-optimal resource allocation.

2. *Impact of Data Density*: High data density causes sub-scaling due to diminishing returns from redundant data. Low-density datasets, with more diverse data, align more closely with traditional scaling laws.

3. *Optimal Resource Allocation*: Efficient computational resource allocation is crucial to mitigate sub-scaling and sustain performance improvements.

4. *Sub-Optimal Scaling Law*: We proposed a Sub-Optimal Scaling Law that generalizes the Chinchilla scaling law to better predict performance and loss in sub-scaling regimes. This new framework accounts for the quality and diversity of training data, emphasizing that these factors often outweigh the benefits of mere scale.

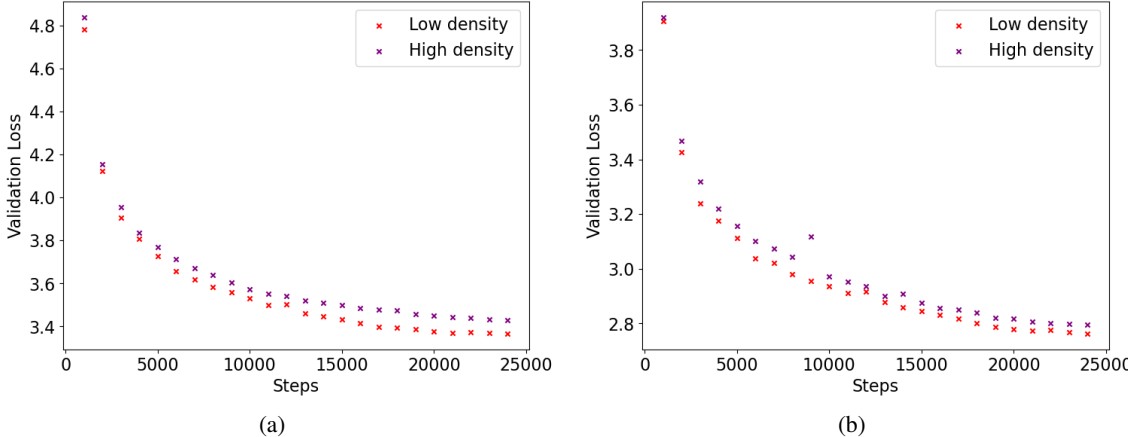

(a)                                                                (b)

Figure 4: We compare models with (a) 100M parameters and (b) 800M parameters trained on high and low density dataset, which demonstrates that higher density results in a degressive performance increase.

## 2 ANALYSIS

In this section, we investigate the phenomenon of sub-scaling law from two perspectives: data density and training strategy. Our analysis is based on extensive empirical evaluations involving over 400 models, ranging from 20 million to 7 billion parameters, trained on various datasets with different training strategies.

### 2.1 ANALYSIS 1: THE PERSPECTIVE OF DATA

Previous works Abbas et al. (2024); Sachdeva et al. (2024); Sorscher et al. (2022) have used data density to measure the quality and diversity of datasets. By focusing on density, these methods aim to identify and remove redundant data points within high-density clusters, effectively reducing redundancy and ensuring a wide range of topics, genres, and linguistic structures are covered. As shown in Figure 4, we observed that high data density often leads to diminishing returns in performance improvements. This phenomenon can be understood through the lens of Information Theory, where redundant information in high-density datasets contributes less new information to the model, thus reducing the marginal gains from additional data.

Figure 5 provides a schematic representation of datasets with different densities. The yellow circle represents a cluster with a large number of samples and high density, where the samples are more similar to each other. In contrast, the gray circle, containing fewer samples and lower density, signifies a cluster where the samples are less similar to each other. This figure illustrates that high-density datasets contain redundant information, which leads to diminishing marginal returns in performance improvements. As a result, the gains from additional data decrease, contributing to the sub-scaling law phenomenon. Analysis of real-world Common Crawl dataset Common Crawl (2024); Gao et al. (2020) is shown in Figure 6. It presents the relationship between samples and clusters, illustrating the differences between the original dataset and the deduplicated low-density dataset. The deduplication process reduces data redundancy, thereby increasing the diversity of the dataset. To evaluate density, different from previous works Sachdeva et al. (2024); Abbas et al. (2024), we propose a density metric that considers both the concentration of samples within individual clusters and the separation between different clusters. This approach aims to provide a more comprehensive measure of data density by accounting for the internal structure of clusters as well as the overall distribution of the dataset.

**Density of a single cluster**. Density of a single cluster reflects the concentration of samples within a cluster. For each cluster $C_i$, its density $\rho_i$ can be defined as the ratio of the number of samples within the cluster to its volumn. Suppose cluster $i$ has $N_i$ samples in $R^n$, and the average radius of the cluster $r_i$ is:

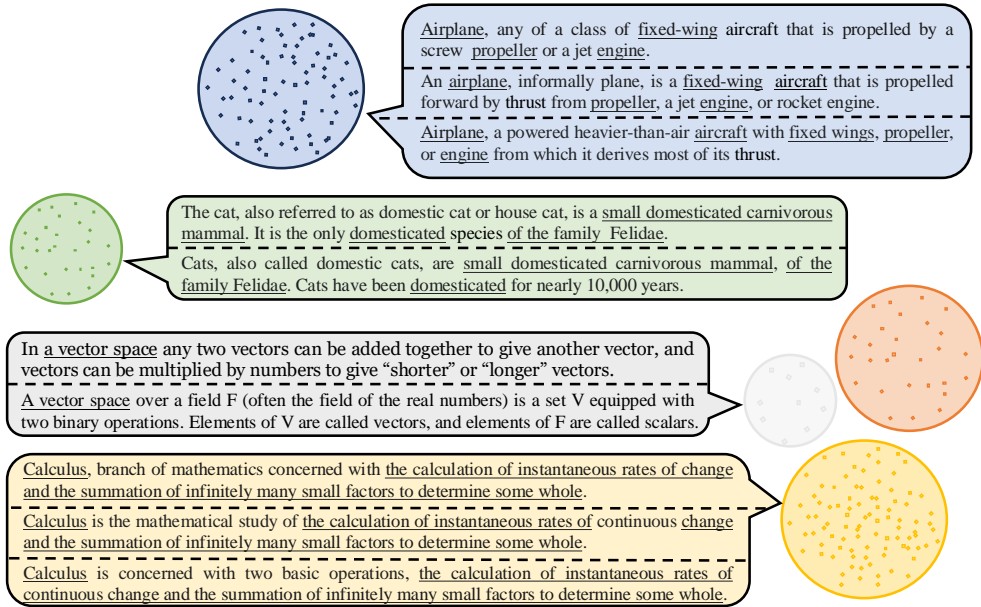

Figure 5: The yellow circle represents a cluster with a larger number of samples and higher density, where the samples are more similar to each other. In contrast, the gray circle, containing fewer samples and lower density, signifies a cluster where the samples are less similar to each other. Considering the relationship between clusters, the yellow, gray, and orange circles are closer to each other, indicating a similar topic (math). On the other hand, the blue and green circles represent clusters with less similarity.

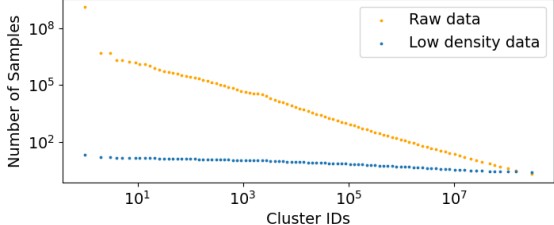

Figure 6: The relationship between number of samples and cluster ID in Common Crawl dataset, with cluster IDs sorted in descending order by the number of samples. Points are sampled for illustration. In the figure, "Raw data" refers to the original dataset, while "Low density data" refers to the dataset after deduplication.

$r_i = \frac{1}{N_i} \sum_{x \in C_i} ||x, c_i||$, where $||x, c_i||$ denotes the distance between sample $x$ and the center $c_k$ of cluster $K$. Then for data with $n$ dimension, the density $\rho_i$ of cluster $i$ can be defined as: $\rho_i = \frac{N_i}{\frac{\pi^{n/2}}{\Gamma(n/2+1)} r_i^n} = \frac{N_i \Gamma(n/2+1)}{\pi^{n/2} r_i^n}$.

**Density of dataset**. Density of dataset reflects the degree of separation between different clusters. For cluster $i$ and its nearest neighbor cluster $j$, suppose the centroid distance between cluster $i$ and cluster $j$ is $||c_i, c_j||$. To introduce the weight of density of single cluster, the weighted radius of the dataset $R$ can be defined as:

$$R = \frac{1}{K} \sum_{i=1}^{K} \frac{||c, c_i||}{log(\rho_i + 1)}, \text{ where } c = \frac{1}{K} \sum_{i=1}^{K} c_i, \tag{1}$$

where $K$ is the total number of classes. We define density of the dataset $\rho$ as:

$$\rho = \frac{N}{\frac{\pi^{n/2}}{\Gamma(n/2+1)} R^n} = \frac{N\Gamma(n/2+1)}{\pi^{n/2} R^n}. \tag{2}$$

where $N$ is the total number of samples in the dataset. Similar to previous works Sachdeva et al. (2024); Abbas et al. (2024), we use density metric to select data, detailed in the Appendix.

In information theory, the amount of new information (or entropy) introduced by additional data plays a critical role in model learning Sachdeva et al. (2024); Abbas et al. (2024). High-density datasets often contain redundant information, while low-density datasets provide more diverse and unique data points. This distinction directly impacts the law governing performance improvement: whether it follows a scaling law or a sub-scaling law.

Low-density datasets, which contain more diverse and less redundant information, exhibit a more linear relationship between the number of samples and performance improvement. Each new sample provides relatively new information, maintaining a steady rate of performance gain. The information gain $I(n)$ for low-density datasets can be expressed as a linear function: $I(n) = I_0 \cdot n$ where $I_0$ is a constant representing the information gain per sample. The performance $P(n)$ in a low-density dataset context is given by: $P(n) = P_0 \left(1 - e^{-\beta \cdot I(n)}\right)$. Given the linear information gain, this simplifies to: $P(n) = P_0 \left(1 - e^{-\beta \cdot I_0 \cdot n}\right)$. For smaller $n$, this can be approximated as a linear function:

$$P(n) \approx P_0 \cdot \beta \cdot I_0 \cdot n. \tag{3}$$

In contrast, high-density datasets, characterized by a significant level of redundancy, tend to exhibit diminishing returns as more data is added. Redundant information increases the mutual information between samples, leading to a lower incremental gain from each additional sample, which leads to the *sub-scaling law*.

The information gain $I(n)$ for high-density datasets can be modeled as: $I(n) = I_0 \cdot n^{-\alpha}$ where $I_0$ represents the initial information gain, and $\alpha$ is a positive constant indicating the rate of diminishing returns. This formulation suggests that as the number of samples $n$ increases, the additional information gained decreases.

The performance $P(n)$ as a function of the number of samples $n$ follows: $P(n) = P_0 \left(1 - e^{-\beta \cdot I(n)}\right)$. Substituting the information gain:

$$P(n) = P_0 \left(1 - e^{-\beta \cdot I_0 \cdot n^{-\alpha}}\right). \tag{4}$$

This equation captures the power-law growth where performance improvements slow down as data density increases leading to sub-scaling law. We substantiated these theoretical models through extensive empirical analysis in the next section.

## 2.2 ANALYSIS 2: THE PERSPECTIVE OF TRAINING STRATEGY

In training LLMs, the training strategy is crucial for enhancing model performance. The training strategy includes model/data allocation and the selection of hyper-parameters such as batch size and learning rate. These factors directly influence the sub-scaling law and the final performance of the model.

Hyper-parameters are critical factors controlling the learning process of the model, with batch size and learning rate being the most important. However, their selection does not significantly impact sub-scaling

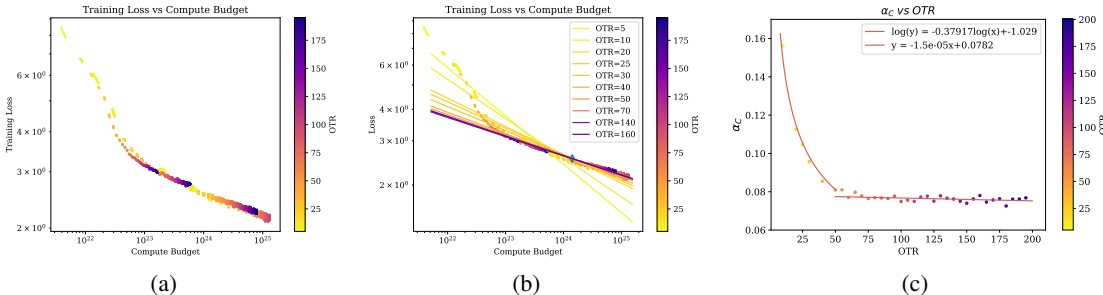

(a)                (b)                (c)

Figure 7: (a) Loss vs. Compute Budget Across Model Sizes. (b) Power-Law Relationship and Convergence Point. (c) For $L = \frac{\lambda_C}{C^{\alpha_C}}$, when $OTR \leq 50$, it is observed that $\alpha_C$ has a power-law relationship with $OTR$. And when $OTR > 50$, $\alpha_C$ remains constant about 0.0782.

phenomena. Non-optimal hyper-parameters tend to show *poor performance from the beginning of the training process rather than causing a deceleration in performance improvement with the dataset or model size increases*, and thus do not lead to the sub-scaling effect, which is detailed in the Appendix C.

Optimal allocation of computational resources between model size and training data is paramount for achieving the best performance while avoiding sub-scaling effects. Striking the right balance ensures that computational resources are used most efficiently, thereby maximizing performance improvements and mitigating the diminishing returns associated with sub-scaling laws. We delve into the impact of different allocation strategies, focusing on the effects of non-optimal allocation especially on over-training (i.e., using more tokens than optimal relative to the model size).

The theoretical foundation for optimal resource allocation is encapsulated by the compute budget relationship: FLOPs$(N, D) \approx 6ND$, where $N$ is the model size (number of parameters), $D$ is the number of training tokens, and the compute budget (FLOPs) is kept constant. The challenge lies in distributing this budget optimally between increasing the model size and augmenting the dataset.

Over-training occurs when the number of training tokens significantly exceeds the optimal allocation for a given model size. This imbalance results in diminishing returns and a divergence from the expected performance improvements based on traditional scaling laws. We quantify over-training using the Over-Training Ratio (OTR): $\text{OTR} = \frac{D}{D_{\text{opt}}}$ where $D$ is the total number of training tokens, and $D_{\text{opt}}$ represents the optimal number of tokens required for the model size $N$.

Figure 7 shows the Over-Training phenomenon, where non-optimal allocation manifests as over-training. In Figure 7 (a) and (b), it is evident that as the OTR increases, the rate of performance improvement decreases, indicating sub-scaling behavior. The diminishing returns are more pronounced when the OTR exceeds a certain threshold, underscoring the critical role of optimal resource allocation in maintaining efficient scaling. In Figure 7 (c), we observed that when the OTR exceeds a threshold of 50, the performance improvements decelerate markedly. This threshold serves as a critical point, beyond which additional data does not contribute efficiently to performance gains, thus indicating an over-trained model.

Previous research (Hoffmann et al., 2022; DeepSeek-AI et al., 2024) indicates that $D_{\text{opt}}$ is positively correlated with the model size $N$, suggesting that larger models generally require more data to reach their optimal training point before over-training begins. Based on this relationship, we refine the definition of OTR to directly relate the amount of data used to the model size, as follows:

$$\text{OTR} = \frac{D}{N}. \tag{5}$$



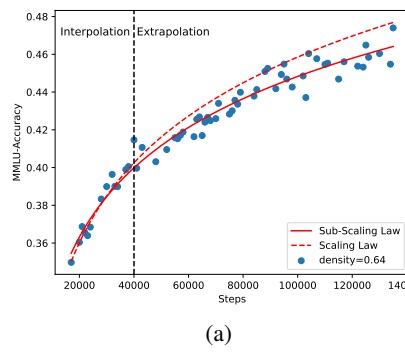
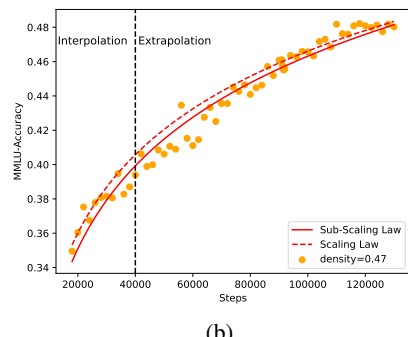

(a)                                                              (b)

Figure 8: The figure illustrates the performance of language models as a function of the number of samples, comparing the traditional scaling law and our proposed sub-optimal scaling law. (a) The traditional scaling law, which assumes a straightforward relationship often leading to power-law growth, does not fit well in high-density datasets due to redundant information reducing marginal gains. Consequently, it shows larger fitting errors in such datasets. (b) In contrast, the sub-optimal scaling law adapts to both high-density and low-density datasets, accounting for diminishing returns in high-density scenarios, and thus provides a more accurate fit across various data densities.

Effective allocation of computational resources is critical to leveraging scaling laws for optimal model performance. Over-training, characterized by an excessive OTR, leads to sub-scaling effects where performance gains diminish rapidly.

## 3 APPROACH

### 3.1 APPROACH 1: ESTIMATING SUB-OPTIMAL SCALING LAW VIA DATA DENSITY

To understand how data density impacts performance, we estimated the power-law relationship for datasets with similar densities. The performance $P(n)$ was modeled using:

$$P(n) = P_0 \left( 1 - e^{-\beta I(n)} \right). \tag{6}$$

For high-density datasets where information gain diminishes, we model the information gain $I(n)$ as: $I(n) = I_0 \cdot n^{-\alpha}$. In high-density datasets, redundancy is prevalent. This means that as more data is added, the amount of new, unique information gained decreases. This redundancy leads to diminishing returns, where each additional sample contributes less to the overall performance improvement of the model. The decay factor $R_D$ allows the model to more accurately represent this non-linear relationship and the reduced effectiveness of additional data. Thus, we use decay factors $R_D$ to show the influence of data density for fitting *sub-optimal scaling*:

$$P = R_D \cdot \lambda \cdot C^\alpha, \tag{7}$$

where $\lambda$ and $\alpha$ are empirically determined coefficients. Given this equation, we could fit different density datasets with varying $R_D$ values to account for the influence of data density on performance. The fitting results are shown in Figure 8.

We train 7B language models with different densities, two different fitting curves are shown in Figure 8: the traditional scaling lawYang et al. (2024) $P = \lambda \cdot C^\alpha$ and our proposed sub-optimal scaling law. The traditional scaling law assumes a more straightforward relationship between the number of training samples and model performance, often leading to a power-law growth. However, this assumption does not hold well in high-density datasets where redundant information reduces the marginal gains from additional data. As

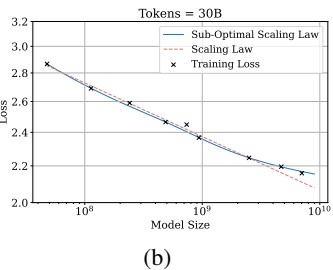 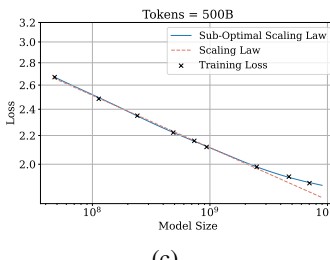

(a)                              (b)                              (c)

Figure 9: We compare the sub-optimal scaling law $L = \frac{\lambda_D \cdot R_D}{D^{\alpha_D}} + \frac{\lambda_N \cdot R_N}{N^{\alpha_N}} + E$ with traditional scaling law Hoffmann et al. (2022) $L = \frac{\lambda_D}{D^{\alpha_D}} + \frac{\lambda_N}{N^{\alpha_N}} + E$ across the same range of training tokens and model sizes. With the number of training tokens increasing, the $R_D, R_N$ becomes larger and gives the curve a larger degree of curvature, which makes a better regression.

a result, the traditional scaling law shows larger fitting errors in high-density datasets, failing to capture the complex dynamics of data redundancy. The sub-optimal scaling law adapts to the characteristics of the dataset, whether it is high-density or low-density. It accounts for the diminishing returns in performance improvements that occur in high-density datasets due to redundant information. This adaptability allows the sub-optimal scaling law to fit a wide range of datasets more accurately.

By incorporating the decay factor $R_D$, our sub-optimal scaling law provides a more nuanced understanding of how data density affects model performance. This approach ensures that the fitting process is sensitive to the density characteristics of the dataset, leading to more accurate performance predictions.

### 3.2 APPROACH 2: PARAMETRIC FIT FOR PERFORMANCE FORMULATIONS

Table 1: Details about the fitting result of traditional scaling law and sub-optimal scaling law.

| Training Tokens (billion) | Fitting MAPE | | Prediction MAPE | |
|---|---|---|---|---|
| | Scaling Law | Sub-Optimal Scaling Law | Scaling Law | Sub-Optimal Scaling Law |
| 5 | 0.00369 | **0.00224** | **0.00757** | 0.00887 |
| 30 | 0.00543 | **0.00328** | 0.01151 | **0.00249** |
| 500 | 0.00296 | **0.00246** | 0.02451 | **0.00156** |

Figure 9 compares the sub-optimal scaling law with traditional scaling laws across the same range of training tokens and model sizes. The results indicate that non-optimal allocation leads to a greater degree of curvature in the loss curve. Table 1 shows the details about the fitting result of traditional scaling law and sub-optimal scaling law, where our proposed sub-optimal scaling law outperform traditional scaling law. This comparison clearly shows that traditional scaling laws fall short in non-optimal allocation scenarios, where non-optimal allocation exacerbates sub-scaling effects.

Figure 10 examines the impact of model/data allocation ratio on training loss under a fixed compute budget. The results demonstrate that non-optimal allocation, where the number of training tokens significantly exceeds the optimal amount, results in a marked increase in training loss. This indicates that improper allocation not only leads to inefficiency but also to a degradation in model performance, further reinforcing the sub-scaling.

We developed parametric models to fit the loss and performance data observed during our experiments. These models account for the over-training effects by incorporating repetition factors $R_N$ and $R_D$ for parameters

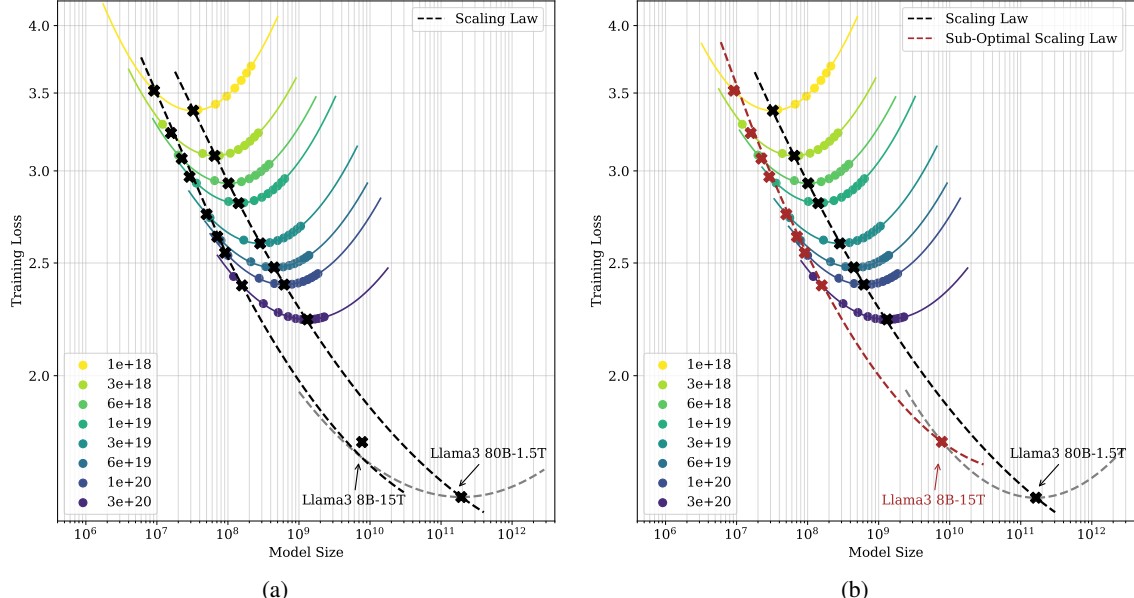

Figure 10: (a) With the total amount of compute budget fixed, we change the model/data allocation ratio, then plot the training loss vs model size curve. We use a black curve to connect the training data of smaller models and do the extrapolation. However, it could be observed that when it comes to the Sub-Scaling situation, the prediction accuracy decreased. The MAPE error is 0.0300. (b) Therefore, considering it is common that the number of training tokens has greatly exceeded the optimal one, we tend to investigate the Sub-Optimal Scaling Law (red line). From the diagram, we could conclude that the Sub-Optimal Scaling Law could capture the performance degradation. The MAPE error is 0.0013.

and data, respectively:

$$L(N, D) = E + \frac{\lambda_N \cdot R_N}{N^{\alpha_N}} + \frac{\lambda_D \cdot R_D}{D^{\alpha_D}}, \tag{8}$$

where $E$ represents the baseline loss, and $\lambda_N, \lambda_D, \alpha_N, \alpha_D$ are empirically determined coefficients. The logistic functions $R_D$ and $R_N$, which control the effects of $D$ and $N$ on $A$ and $B$, respectively, which is detailed in Appendix. Our findings highlighted the critical role of optimal resource allocation and data quality, providing a robust framework for predicting performance in sub-scaling regimes.

## 4 CONCLUSION

This study systematically investigated the sub-scaling law phenomenon in large language models (LLMs), where performance improvements decelerate as model or dataset size increases. Through extensive empirical analysis of over 400 models, we identified key factors contributing to sub-scaling: high data density and non-optimal training resource allocations. Our findings highlight that high data density leads to diminishing marginal gains in performance, while optimal resource allocation is crucial for sustaining improvements. We proposed a Sub-Optimal Scaling Law that better predicts performance and loss in sub-scaling regimes by accounting for data quality and diversity. Future research should refine this sub-optimal scaling law and explore its applicability to other models and tasks.

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

Table 2: Density of Datasets.

| Dataset | The Pile | Deduplicated Pile | Density-Based Pile |
|---------|----------|-------------------|--------------------|
| Density | 0.64 | 0.56 | 0.47 |

Greg Yang, Dingli Yu, Chen Zhu, and Soufiane Hayou. Tensor programs vi: Feature learning in infinite-depth neural networks. *arXiv preprint arXiv:2310.02244*, 2023.

## A  EXPERIMENTAL SETTINGS

To evaluate the impact of over-training on the performance of language models, we designed a comprehensive experimental framework. This section outlines the models, datasets, and training setup, used to assess the effects of over-training and validate our proposed scaling laws. Our code and data are released in https://github.com/AnonymousCode222/SOSL.

### A.1  DATASET

For all datasets utilized in our study, we adhered to a systematic preprocessing routine to ensure consistency and optimize training efficiency similar to previous works (Xie et al., 2024; Hoffmann et al., 2022).

To enhance the training process, we organized the examples by domain. This organization allows for hierarchical sampling, where a domain is first selected according to predefined weights, and subsequently, an example is randomly chosen from the selected domain. This method ensures that the model is exposed to a diverse range of text styles and content during training, which is critical for developing robust language models.

Furthermore, to reduce the number of padding tokens, which can adversely affect model training efficiency and performance, we employed a packing strategy. In this strategy, examples, potentially from different domains, are packed together into the same sequence. This approach not only reduces padding but also introduces a degree of domain variability within the same batch, potentially enhancing the model's ability to generalize across different types of text.

The primary dataset used in our experiments is The Pile (Gao et al., 2020), a comprehensive text corpus that consists of approximately 800GB of text spanning 22 different domains. The Pile is well-regarded in the language modeling community for its diversity and volume, making it an ideal choice for studying the effects of over-training. The default sampling weights for The Pile were determined based on heuristic methods to balance the representation of each domain adequately.

By meticulously preparing and preprocessing our datasets, we ensure that the models are trained under optimal conditions, allowing us to accurately assess the effects of over-training and the validity of the newly proposed scaling laws.

**Density of Datasets**. In our study, we utilized a proposed density calculation formula Eq.2 to evaluate the density of various datasets. The density of a dataset is a critical factor influencing the performance and scaling behavior of large language models. Higher density indicates more redundancy and less diversity in the data, which can lead to sub-scaling phenomena.

The table 2 presents the density values for three different datasets: The Pile, Deduplicated Pile, and Density-Based Pile.

- The Pile: This dataset has a density of 0.64, indicating a relatively high level of redundancy and less diversity.
- Deduplicated Pile: After removing redundant data points, the density of this dataset is reduced to 0.56, showing an improvement in data diversity.
- Density-Based Pile: Similar to previous works Sachdeva et al. (2024); Abbas et al. (2024), we use metric density to select data from The Pile dataset to get the Density-Based Pile, which is used as a low-density dataset in our paper. This dataset is curated based on density considerations, resulting in a lower density of 0.47, which indicates higher diversity and less redundancy.

These values highlight the differences in data redundancy and diversity across the datasets. Lower density values suggest datasets with more diverse and unique data points, which are crucial for mitigating sub-scaling phenomena and achieving better performance in large language models.

## A.2 MODEL ARCHITECTURE

The architecture details of our models involved in all experiments are presented in Table 3. Each model configuration varies in terms of size, hidden layers, head size, feedforward network (FFN), etc.

Table 3: Details about the architecture of our models involved in all experiments.

| Parameters (million) | Hidden Size | Layers | Head Size | FFN |
| --- | --- | --- | --- | --- |
| 20 | 384 | 12 | 4 | 960 |
| 47 | 512 | 14 | 4 | 1536 |
| 113 | 768 | 16 | 6 | 2048 |
| 241 | 1024 | 20 | 8 | 2560 |
| 487 | 1280 | 24 | 10 | 3584 |
| 736 | 1536 | 26 | 12 | 4096 |
| 936 | 1664 | 28 | 13 | 4480 |
| 1330 | 1920 | 30 | 16 | 5120 |
| 2510 | 2560 | 32 | 20 | 6784 |
| 4700 | 3328 | 32 | 26 | 8864 |
| 7030 | 4096 | 32 | 32 | 11008 |

## A.3 TRAINING SETUP

All models were trained using the AdamW optimizer with $\beta_1 = 0.9$ and $\beta_2 = 0.95$. We followed the Chinchilla law to determine the maximum learning rate, setting it at $2 \times 10^{-4}$ for smaller models and $1.25 \times 10^{-4}$ for larger models. A cosine scheduler with a 10x learning rate decay was employed during training (Hoffmann et al., 2022) . To ensure optimal training without sub-optimal models, we used Gaussian smoothing with a 10-step window length to refine the training curve.

The specific range of hyperparameter settings, including batch size and learning rate, were tailored to each model size to achieve optimal performance within the allocated FLOP budget.

The key variable in our experiments was the over-training ratio, defined as the ratio of the number of training tokens and model size.

## B  DENSITY'S IMPACT ON PERFORMANCE GROWTH: POWER-LAW VS. LINEAR

To understand how the density of data affects performance growth patterns—whether it follows a power-law or a linear function—we can look at it from the perspectives of Information Theory. Here, we'll explore potential formulas and theoretical explanations for these phenomena.

**Information Theory Perspective**. From Information Theory, we can quantify the amount of new information (or entropy) gained from each additional sample. The key concept here is the redundancy of information in high-density datasets versus the diversity in low-density datasets.

**High-Density Datasets (Power-Law Growth)**.

Redundant Information: When a dataset has many duplicates, the mutual information between samples increases, leading to redundancy. The additional information gained from each new sample decreases as the number of samples increases.

Diminishing Returns: The concept of diminishing returns can be described by a power-law function. If $I(n)$ represents the information gained after seeing $n$ samples, it can be modeled as:

$$I(n) = I_0 \cdot n^{-\alpha}$$

where $I_0$ is a constant representing the initial information gain and $\alpha$ is a positive constant indicating the rate of diminishing returns.

Performance Growth: - Performance $P(n)$ as a function of the number of samples $n$ can be expressed as:

$$P(n) = P_0 \cdot (1 - e^{-\beta \cdot I(n)})$$

where $P_0$ is the maximum achievable performance and $\beta$ is a constant scaling the information gain to performance.

**Low-Density Datasets (Linear Growth)**

Diverse Information: In low-density datasets, each sample provides more unique information, leading to a more consistent information gain.

Consistent Returns: The information gain in a low-density dataset can be approximated as a linear function:

$$I(n) = I_0 \cdot n$$

where $I_0$ is a constant representing the information gain per sample.

Performance Growth: Performance $P(n)$ in a low-density dataset can be modeled as:

$$P(n) = P_0 \cdot (1 - e^{-\beta \cdot I(n)})$$

Given $I(n)$ is linear, this simplifies to:

$$P(n) = P_0 \cdot (1 - e^{-\beta \cdot I_0 \cdot n})$$

For small $n$, this can be approximated as a linear function:

$$P(n) \approx P_0 \cdot \beta \cdot I_0 \cdot n$$

High-Density Datasets: Performance growth follows a power-law due to diminishing returns from redundant information. This can be modeled using:

$$P(n) = P_0 \cdot (1 - e^{-\beta \cdot n^{-\alpha}})$$

Low-Density Datasets: Performance growth is more linear due to consistent returns from diverse information. This can be modeled using:

$$P(n) = P_0 \cdot \beta \cdot I_0 \cdot n$$

These models explain why high-density datasets lead to power-law growth in performance, while low-density datasets result in linear performance growth.

## C  SCALABLE HYPER-PARAMETERS FOR SUB-SCALING LAW

While the selection of hyper-parameters such as batch size and learning rate is crucial for the overall training efficiency and effectiveness, it does not significantly impact sub-scaling phenomena. Non-optimal hyper-parameters show poor performance from the beginning of the training process, rather than causing a deceleration in performance improvement as the dataset or model size increases. Our findings demonstrate that both batch size and learning rate can be optimized in a scalable manner that is robust to over-training. The power-law relationships established through empirical data provide a solid foundation for setting these hyper-parameters efficiently across different model sizes and training conditions. By optimizing these hyper-parameters, we can achieve better performance while minimizing the computational expenditure and mitigating the effects of over-training.

In the context of training large language models, managing the balance between sufficient training and over-training is crucial for achieving optimal performance. Over-training can lead to diminished returns and wasted computational resources. To address these challenges, we propose an optimal hyper-parameter search method that is robust to the effects of over-training.

Hyper-parameters play a critical role in the training of machine learning models. They control the learning process and significantly influence the model's performance. However, the traditional hyper-parameter optimization process does not take into account the potential for over-training, leading to selections that may not be optimal under over-training conditions. Our proposed method focuses on identifying hyper-parameters whose performance relationship remains stable and unaffected by over-training. This stability ensures that the chosen hyper-parameters continue to yield the best possible performance even as the amount of training data significantly exceeds the optimal level.

In the pursuit of optimizing training strategies for large language models (LLMs), understanding the interplay between hyper-parameters and over-training is crucial. Our empirical studies focus on the relationship between the number of processed tokens and key hyper-parameters like learning rate and batch size, across various model sizes and over-training ratios (OTR). This analysis helps in identifying scalable hyper-parameters that maintain efficiency and performance even in over-training scenarios. Our experiments were conducted on models of different sizes: 50M, 100M, 500M, and 1B parameters. The primary goal is to determine the optimal batch size and learning rate that achieve a specific training loss with the minimal number of processed tokens.

**Scaling Strategy for Batch Size** Following the methodology proposed by previous works Kaplan et al. (2020); Hu et al. (2024), we conducted experiments to identify the optimal batch size for models of various scales under over-training. In all experiments, the batch size is counted in sequences with length of 8196. We explore the relationship of optimal batch size and training loss at fixed learning rates ranging from 1e-5 to 1e-3. Figure 11 shows how we explore the relationship with batch size at fixed learning rates of 2.5e-4 and 1.5e-3. We display the distribution heatmap of the loss value in relation to the tokens processed and batch sizes. The red solid vertical lines in these figures mark the minimum number of tokens required to reach a predetermined training loss, highlighting the optimal batch size for efficiency.

Further analysis involved connecting the minima lines from the 4 model sizes, as shown in Figure 12, the minima of these parabolas, connected by red lines, indicate the shifts in optimal batch size as the loss

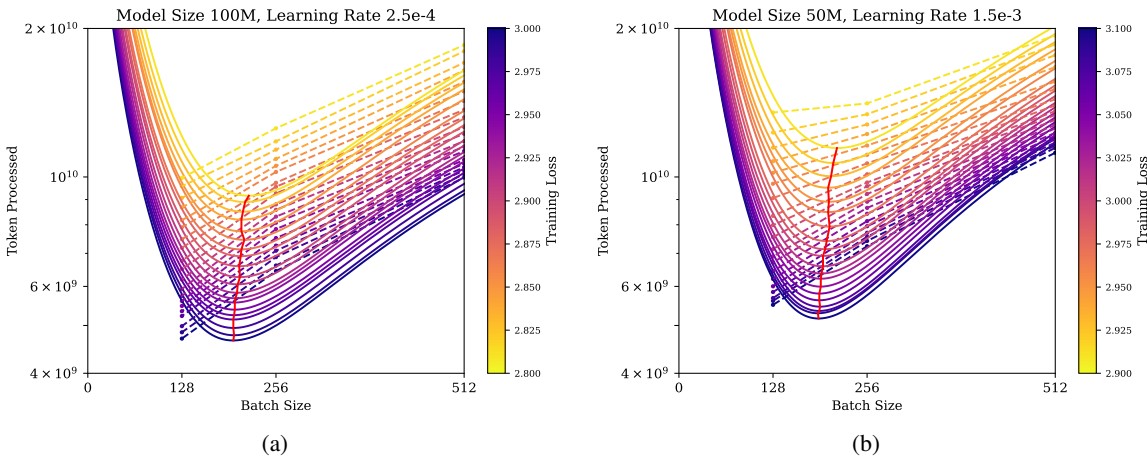

Figure 11: The number of processed tokens relative to different batch sizes needed to reach a specific training loss, under two distinct learning rate conditions. (a) displays this relationship at a learning rate of 2.5e-4, and (b) at a learning rate of 1.5e-3.

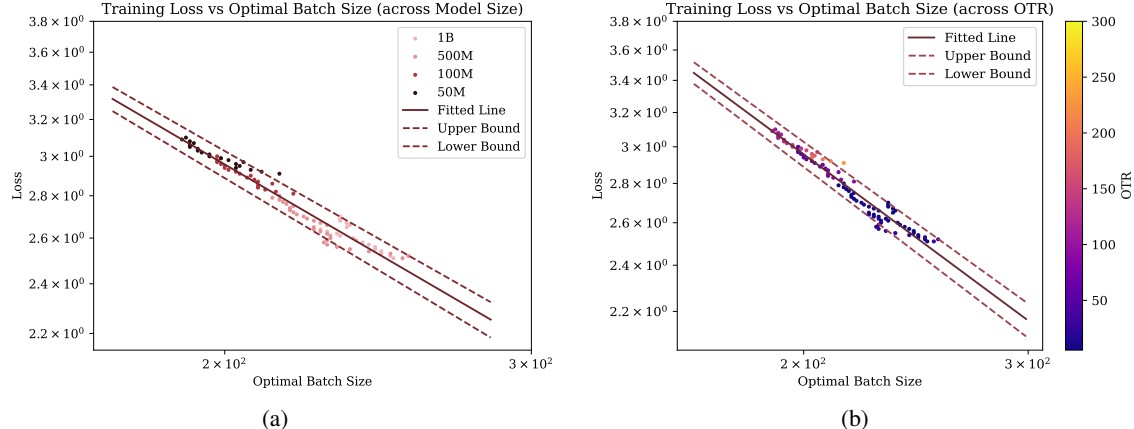

Figure 12: The empirical data points show a scattered distribution around the fitted power-law line for loss and the optimal batch size. This diagram illustrates that optimal batch size values scale proportionally with the loss value and remain unaffected by changes in model size. The scattered dots represent empirical results of optimal batch size and training loss, with the solid line in the center indicating the fitted line. Additionally, dotted lines denote the upper and lower bounds for the distribution of scattered dots. (a) showcases the optimal batch size versus loss with different model sizes, while (b) displays the optimal batch size versus loss with different over-training ratio (OTR). Despite the variations of model size and OTR, all points consistently align around the same curve, highlighting the robustness and universality of the observed power-law relationship.

decreases. This method revealed that as the loss diminishes, the optimal batch size increases, suggesting a dynamic relationship between batch size and dynamic loss values.

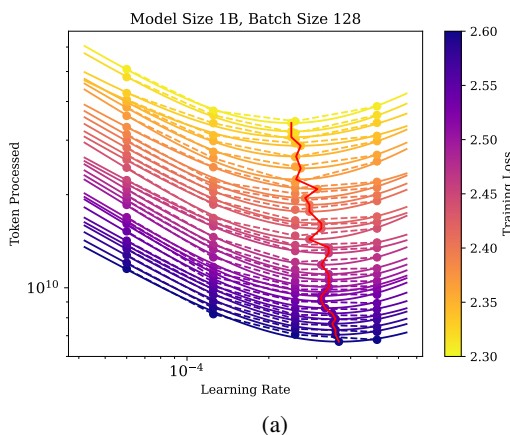 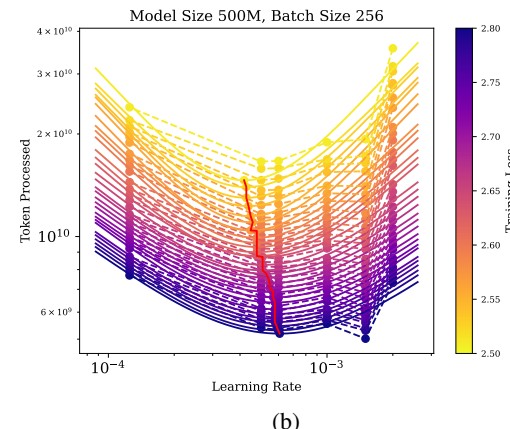

(a)  (b)

Figure 13: The relationship between tokens and learning rate required to achieve a specific training loss across four different model sizes. The red solid vertical line highlights the minimum number of tokens needed at the optimal learning rate for a predetermined training loss. (a) explores this relationship when the batch size is set to 128, while (b) examines the scenario with batch size 256.

From this relationship, we derived the following formula that relates batch size to the loss:

$$\hat{L}(B) = \lambda_B \cdot B^{-\alpha_B} \tag{9}$$

where the $\lambda_B, \alpha_B$ are coefficients and $L, B$ are loss value and optimal batch size at a specific loss value. This equation describes how the loss varies inversely with the batch size, following a power-law behavior. Moreover, such a relationship shown in Equation 9 remains consistent across different model sizes and OTRs. This consistency underscores the robustness of the batch size as a scalable hyper-parameter in the face of over-training.

**Scaling Strategy for Learning rate** We extended methodologies from prior research Kaplan et al. (2020); Hu et al. (2024), where we also conduct experiments towards identifying the optimal learning rate that balances computational efficiency and model performance effectively.

We explored the relationship between learning rate and training loss at fixed batch sizes raning from 128 to 2048, as shown in Figure 13. In this figure, we show the relationship of training loss with the optimal learning rate and tokens processed. The red solid vertical lines mark the minimum number of tokens required to achieve a specified training loss, pinpointing the optimal learning rate for efficiency.

As shown in Figure 14, the linkage of these minima loss across four different model sizes revealed a linear relationship in logarithmic space, leading to the derivation of the following formula that quantitatively describes the relationship between learning rate and loss:

$$\hat{L}(\eta) = \lambda_\eta \cdot \eta^{-\alpha_\eta} \tag{10}$$

where $L, \eta$ are loss value and optimal leanring rate at a specific loss value, respectively. And $\lambda_\eta, \alpha_\eta$ are coefficients. Equation 10 captures the optimal learning rate's dependency on training loss. We observed a similar power-law relationship between the optimal learning rate and the training loss, irrespective of OTR and model size. The empirical data points, represented in Figure 14, consistently align around the fitted

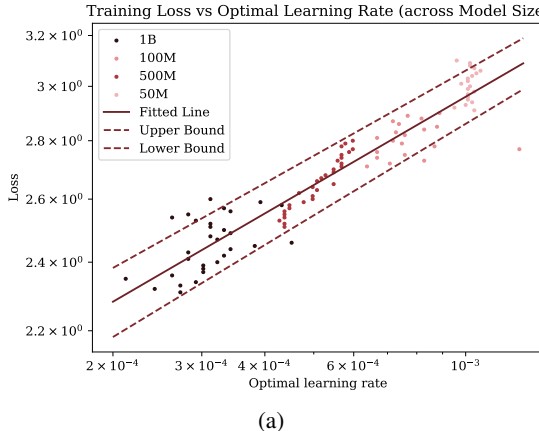 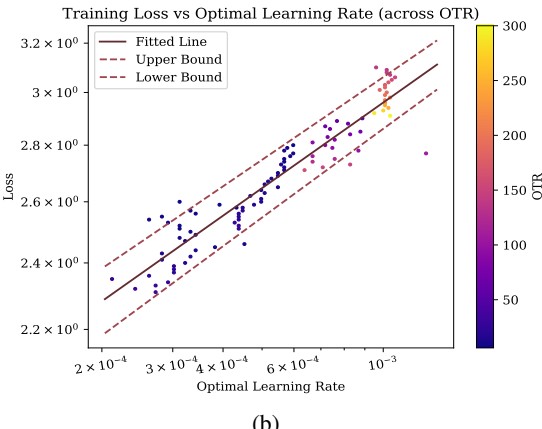

(a)                                              (b)

Figure 14: The empirical data points exhibit a scattered distribution around the fitted power-law line for the loss and critical learning rate relationship. The diagram illustrates that critical learning rate values scale proportionally with the loss value and remain independent of the model size at a specific training loss value. The solid line represents the fitted line, while the dashed lines denote the upper and lower bounds, respectively. (a) showcases the optimal learning rate versus loss with different model sizes, while (b) displays the optimal learning rate versus loss with different Over-Training Ratios (OTR). Despite the variations in model size and OTR, all points consistently align around the same curve, highlighting the robustness and universality of the observed power-law relationship.

power-law lines, indicating that the learning rate scales proportionally with the loss and remains unaffected by changes in model size or OTR at a specific training loss value. This trend is consistent across different model sizes and OTRs, highlighting the robustness and scalability of learning rate as a hyper-parameter in the context of over-training.

Our findings demonstrate that both batch size and learning rate can be optimized in a scalable manner that is robust to over-training. The power-law relationships established through empirical data provide a solid foundation for setting these hyper-parameters efficiently across different model sizes and training conditions. This scalability ensures that models are trained effectively, maximizing performance while minimizing unnecessary computational expenditure due to over-training.

## D    EXTENSION OF FITTING DETAILS

**Details about the optimal allocation strategy validation.** Figure 15 explores the correlation between training loss and compute budget across different levels of OTR values. The training loss is denoted as $L = \lambda_C \cdot C^{-\alpha_C}$, where $L, C$ are training loss value and the computing budget, $\lambda_C$ and $\alpha_C$ are coefficients. The dashed scattered points represent the relationship between training loss and compute budget across various model sizes. A set of lines in the graph indicates that the training loss and compute budget, across different OTR levels, still maintain a power-law relationship. Notably, these lines converge at a fixed point (marked as the green point). Before this point, the training loss decreases rapidly. After OTR = 20, the performance improvement from increasing the OTR diminishes. Beyond the green point, the effect of overtraining becomes significant, and reducing the OTR value can lead to performance gains. Thus, the optimal OTR value should

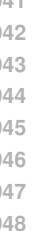

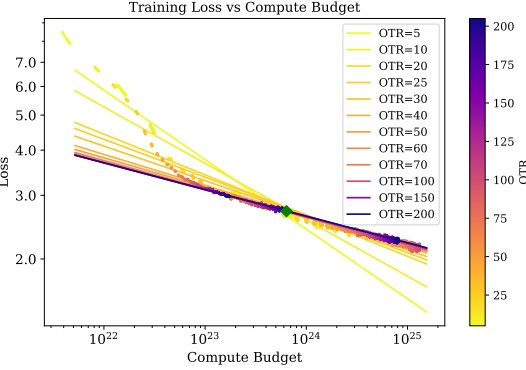

Figure 15: In a log-log diagram, we demonstrate the relationship of training loss and compute budget, with $L = \frac{\lambda_C}{C^{\alpha_C}}$. It could be observed that the value of $\alpha_C$ decreases from $OTR = 5$ to $OTR = 50$, then remains stable.

be around 20, validating the optimal data/model allocation strategy proposed in previous work (Hoffmann et al., 2022).

Table 4: Details about the fitting result of traditional scaling law and sub-optimal scaling law.

| Model Size | Fitting MAPE | | | Prediction MAPE | | |
|---|---|---|---|---|---|---|
| | Sub-Optimal Scaling Law | Hoffman Scaling Law | OpenAI Scaling Law | Sub-Optimal Scaling Law | Hoffman Scaling Law | OpenAI Scaling Law |
| 50M | 0.00099 | 0.00125 | **0.00097** | **0.00207** | 0.00836 | 0.00398 |
| 100M | 0.00193 | 0.00142 | **0.00109** | **0.00430** | 0.00763 | 0.00797 |
| 300M | 0.00225 | 0.00214 | **0.00190** | **0.00346** | 0.01061 | 0.00350 |
| 500M | **0.00101** | 0.00271 | 0.00208 | **0.00079** | 0.00255 | 0.00414 |
| 700M | **0.00068** | 0.00075 | 0.00092 | **0.00086** | 0.00088 | 0.00471 |
| 1B | **0.00260** | 0.01261 | 0.00990 | **0.00267** | 0.00846 | 0.00925 |
| 2B | **0.00754** | 0.01184 | 0.01280 | **0.01045** | 0.01865 | 0.01462 |
| 4B | **0.00354** | 0.00677 | 0.01518 | 0.00335 | **0.00307** | 0.00797 |
| 7B | **0.01207** | 0.01283 | 0.03634 | **0.00254** | 0.00397 | 0.00553 |

**Details about the fitting process of the sub-optimal scaling law.** For LLMs with a model size under 1B, we use the first quarter of the training data to predict subsequent loss values. For larger LLMs (over 1B), the overall range of the OverTraining Ratio (OTR) is narrower due to the computing resources limitation. Therefore, for each specific LLM model, the training data used for fitting includes the loss values and the number of training tokens from all smaller LLMs, along with data from the first quarter of the training phase.

The fitting results are shown in Table 4. The Sub-Optimal Scaling Law outperforms other scaling laws across all model sizes, meaning it better captures performance degradation. As seen in Figure 16, the traditional scaling law proposed by Hoffmann et al. (2022) tends to predict the loss values too optimistically, failing to account for the increasing tendency of $\lambda_D$ and $\lambda_N$. In contrast, the scaling law proposed by Kaplan et al. (2020) sometimes predicts the loss curve too pessimistically.

The final fitting results for the Sub-Optimal Scaling Law:

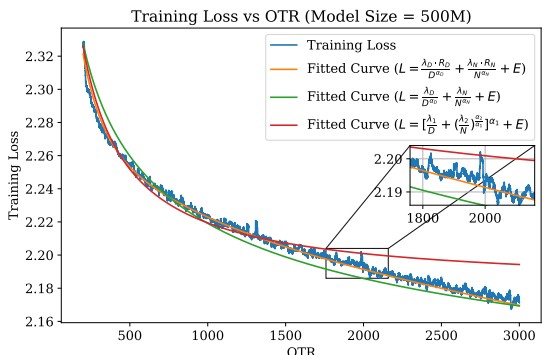

Figure 16: The training data includes the loss values and the number of tokens when $OTR \leq 750$, then we use the hyperparameters obtained to predict the loss curve when $OTR > 750$.

$$L = \frac{455.345 \cdot R_D}{D^{0.289}} + \frac{61.929 \cdot R_N}{N^{0.272}} + 1.372$$

Formula for $R_D$ (Effect of $D$):

$$R_D = 1 + \frac{1}{1 + \exp\left(-k_1 \cdot OTR\right)} \tag{11}$$

Formula for $R_N$ (Effect of $N$):

$$R_N = 1 + \frac{1}{1 + \exp\left(-k_2 \cdot OTR\right)} \tag{12}$$

The logistic functions $R_D$ and $R_N$, which control the effects of $D$ and $N$ on $A$ and $B$, respectively, based on the OverTraining Ratio ( $OTR = \frac{D}{N}$ ). And $k_1, k_2$ are constants which control the steepness of the transition. Our findings highlighted the critical role of optimal resource allocation and data quality, providing a robust framework for predicting performance in sub-scaling regimes.

Specifically,

$$R_D = 1 + \frac{1}{1 + \exp\left(-0.00810 \cdot \text{OTR}\right)}$$

$$R_N = 1 + \frac{1}{1 + \exp\left(-0.00114 \cdot \text{OTR}\right)}$$

with OTR $= \frac{D}{N}$.

## E  RELATED WORKS

**Language Models** Recently, a variety of large language models (LLMs) (Touvron et al., 2023; Hoffmann et al., 2022; Rae et al., 2021; Jiang et al., 2023; Reid et al., 2024; Team et al., 2023; DeepSeek-AI et al., 2024) have emerged, demonstrating excellent performance in the field of language processing. However, during the exploration in this field, the constraints of existing computational resources make training LLMs exhausting. Notably, the proposed scaling law suggests that the performance of smaller models can be extrapolated to

larger ones (Kaplan et al., 2020), highlighting the significance of Small Language Models (SLMs). SLMs are generally defined as models smaller in scale compared to well-known LLMs like Chinchilla (Hoffmann et al., 2022), typically not exceeding 7 billion parameters (Hu et al., 2024). Nonetheless, many factors impact performance improvement for SLMs, such as the availability of training tokens (Muennighoff et al., 2023). In this paper, we focus on the performance of LLMs under over-training conditions.

**Scaling Laws for Large Language Models** The development of large language models (LLMs) has sparked significant interest in understanding their scaling laws due to the huge costs associated with their training. Recent studies (Bahri et al., 2021; Kaplan et al., 2020) suggest a power-law relationship between the loss and the number of non-embedding parameters, dataset size, and compute budget for autoregressive language models (LM) across various scales. However, another study (Hoffmann et al., 2022) adjusted training settings, including training tokens and learning rate schedules, and concluded that model size and training tokens should be scaled equally, contrary to the findings in (Kaplan et al., 2020). Moreover, research (DeepSeek-AI et al., 2024) explores the scaling laws of batch size and learning rate relative to non-embedding FLOPs/token $M$, rather than model scale. This study presents an allocation strategy for scaling up models and data. Additionally, it investigates the impact of pre-training data quality on model performance. Furthermore, their results show that with the same compute budget, the optimal parameter space varies slightly, which has been attributed to the selection of hyperparameters and training dynamics. Recent empirical studies Hernandez et al. (2022a); Hu et al. (2023); Porian et al. (2024); Muennighoff et al. (2024) have observed deviations from this expected trend, particularly in the context of exceptionally large language models. These deviations manifest as sub-scaling growth, where the rate of performance improvement decelerates as model or dataset size continues to increase. Specifically, Hernandez et al. (2022a); Muennighoff et al. (2024) observe that sub-scaling occurs in scenarios involving repeated training data, leading to diminishing returns in performance. Hu et al. (2023) highlight that sub-scaling is particularly pronounced in tasks requiring complex reasoning or multi-step processes. Furthermore, Porian et al. (2024) find that sub-scaling exists under non-optimal training strategies with sub-optimal hyper-parameters.

Compared to recent works (Kaplan et al., 2020; Du et al., 2022; Henighan et al., 2020; Ghorbani et al., 2021; Hernandez et al., 2021), which focus on general scaling laws, our study provides a detailed examination of sub-scaling laws under specific conditions. Previous research (Gadre et al., 2024) has also highlighted sub-scaling under over-training conditions, and some works (Muennighoff et al., 2024; Hernandez et al., 2022b) have explored scaling laws in data-constrained regimes. However, there is a lack of systematic research on the sub-scaling behavior of large language models (LLMs).

**Language Model Behaviour under Over-training** As the computational resources required to train larger models increase, over-training a smaller language model can be a more efficient strategy to achieve performance on par with larger models. It has been observed that smaller models can sometimes outperform larger ones. For instance, the 2B model MiniCPM (Hu et al., 2024) demonstrates capabilities comparable to those of larger models such as Llama2-7B (Touvron et al., 2023) and Mistral-7B. This underscores the importance of Small Language Models (SLMs) (Hu et al., 2024) and the influence of over-training conditions (Gadre et al., 2024). However, there is a lack of thorough investigation into scaling laws under over-training conditions for SLMs. Previous work (Gadre et al., 2024) focuses more on the optimal allocation of tokens and model size rather than its impact on the extrapolation of the scaling law. This work focuses on the performance degradation or improvement caused by over-training and its impacts on the scaling law.

# F   LIMITATIONS

Despite the advancements in over-training scaling laws and scaling strategies for batch size and the robust methodologies employed, our study encompasses several limitations that must be acknowledged: Model Dependency: The results and scaling laws derived from our experiments are based on specific model architectures and configurations. These findings may not universally apply to all types of models, particularly

those with significantly different architectures or training algorithms. Over-Training Focus: While our study provides insights into the effects of over-training, it predominantly focuses on this aspect. Other critical factors such as underfitting, model robustness, and generalization across different tasks and domains were not extensively explored.

## G  BROADER IMPACTS

The implications of our research extend beyond the technical advancements, having several broader impacts: Efficiency in Resource Usage: By optimizing batch sizes and understanding over-training dynamics, our research contributes to more efficient use of computational resources. This not only reduces the cost but also the environmental impact of training large models, aligning with sustainability goals. Accessibility of AI Technology: Improved efficiency and a deeper understanding of training dynamics could lower the barriers to entry for deploying advanced AI models. This could democratize access to AI technologies, allowing a broader range of participants to innovate and develop AI solutions. Ethical Considerations: The ability to train models more efficiently and with better understanding of their limits and capabilities can help in designing more ethical AI by reducing biases and improving fairness. However, the focus on large models might also centralize power among entities that can afford such models, potentially leading to disparities in AI advancements. Educational Valu: Our findings contribute to the academic and practical understanding of machine learning, providing valuable insights for educators, researchers, and practitioners. This can help in curating more effective curricula and training programs that focus on the critical aspects of AI training. Policy and Regulation: As AI technologies become more efficient and widespread, there is a growing need for policies and regulations that ensure these technologies are used responsibly. Our research can inform policymakers about the technical aspects of AI training, aiding in the creation of informed regulations that balance innovation with public welfare.

## H  SAFEGUARDS

In conducting experiments on large language models (LLMs), it is crucial to implement stringent safeguards to ensure the integrity of the research and the ethical use of resources. We have safeguards include: Resource Allocation: Efficiently managing computational resources to minimize environmental impact. This involves optimizing algorithms and models for energy efficiency and prioritizing the use of renewable energy sources when possible. Reproducibility: Maintaining clear documentation of all experimental procedures, configurations, and results to ensure that the experiments are reproducible by other researchers. This transparency is crucial for validating findings and facilitating further research.

## I  COMPUTE RESOURCES FOR EXPERIMENTS

The experiments conducted in this study required substantial computational resources, given the scale and complexity of the LLMs involved. Here is an overview of the compute resources utilized:

Hardware: The experiments were primarily run on high-performance GPU clusters, equipped with the latest graphics processing units capable of handling extensive parallel computations required for training LLMs. Each model size, ranging from 50M to 7B parameters, was allocated appropriate GPU resources to balance efficiency and performance.

Software: We used state-of-the-art machine learning frameworks that support distributed computing. These frameworks were optimized for performance and were instrumental in managing the computational workload efficiently.

