# OpenReview forum: "From Scaling Law to Sub-Scaling Law: Understanding the Diminishing Returns of Larger Models"
_ICLR.cc/2025/Conference — ICLR 2025 Conference Withdrawn Submission_

### Official Review · Reviewer_g7pr · 2024-11-01

**Soundness:** 3
**Presentation:** 3
**Contribution:** 3
**Rating:** 5
**Confidence:** 4

**Summary:**

The authors observe that standard scaling laws have poor fit for the very largest LLMs, with them performing worse than would be expected. They propose that this is because of decreasing marginal information gain in the datasets used by these models and propose 'sub-scaling laws' that account for this limitation, finding that it has better fit to what is observed. The implications of this is that more of the compute budget should be allocated to larger models instead of more tokens.

**Strengths:**

- The paper tries to incorporate the data quality directly into scaling laws, as opposed to past works like Muennighoff et al., which imposes a particular structure (entire repeats of datasets).
- The proposed scaling laws have better fit to observed outcomes.
- The theoretical findings have useful empirical takeaways, namely that existing sota models are likely overtrained.

**Weaknesses:**

- The paper suffers from a lack of rigorous definitions and justification for why certain formalisms have been chosen.
- 229 - 231: Where does this definition of data density come from? Why is information gain measured geometrically instead of say, using the notion of V-information or some other framework?
- The norm that is used to measure distance in embedding space is not explicitly stated. Moreover, LLM embedding spaces tend to be highly anisotropic, yet this metric does not seem to adjust for that at all.
- They decay factor in 3.1 is just another hyperparameter to be tuned; it is not all that surprising that adding another degree of freedom allows the curve to better fit the observed pattern. If the decay factor could be calculated automatically, as a function of the data itself, the result would be much more impressive. In fact, the geometric notion of data density not finding its way into the proposed sub-scaling laws suggests that they are not all that useful as a formalism and are yet another way of some data is lower-quality than other data.

**Questions:**

- The terms in 251 to 273 need to be defined rigorously: is information gain here the reduction in entropy of the next token? What does performance mean, the perplexity? What is $P_0$?

---

> ### Author Response · Authors · 2024-11-27
> **Response to Reviewer g7pr- part 1**
>
> Dear reviewer g7pr, we appreciate the reviewer's perception of our contributions to both empirical and theoretical analysis, and we thank the reviewer for their insightful questions. We believe that there are some important misunderstandings. Please find our detailed responses below:
>
>
>
> **Q1: The paper suffers from a lack of rigorous definitions and justification for why certain formalisms have been chosen. The terms in 251 to 273 need to be defined rigorously: is information gain here the reduction in entropy of the next token? What does performance mean, the perplexity? What is P0?**
>
> **A1:** Thank you for the constructive suggestion.
>
> We will provide definitions for the terms used:
>
> - Information gain: In this context, information gain refers to the reduction in entropy of the next token due to the addition of new data points. This reflects the model's ability to learn from new information.
>
> - Performance: Performance is primarily measured by the model's downstream performance. For example, in Figure 8, we predict the MMLU accuracy as downstream performance.
>
> - P0: P0 represents the maximum achievable performance of the model in its initial state, assuming no data redundancy and optimal conditions.
>
> Moreover, we add the explanations of our proposed data density.
> We will add these definitions to the paper and ensure consistent and clear use of all terms.
>
>
>
> **Q2: 229 - 231: Where does this definition of data density come from? Why is information gain measured geometrically instead of say, using the notion of V-information or some other framework?**
>
>
> **A2:** Thank you very much for your insightful comments!
>
> Previous works [1,2,3] have used data density to measure the quality and diversity of datasets.
> Our definition of \(r_i\) describes the diversity. A larger \(r_i\) indicates greater diversity (of a cluster). The combination with the number of samples comes from the original concept of density: mass/volume. Here, "mass" is the number of samples divided by volume. The geometric measurement of volume is indeed a topic worthy of deeper exploration, which we will explore in further works. The relationship between multiple clusters is a more macro-level calculation. We consider each cluster as a single entity, and its original density gives it a certain weight, thus defining the overall density.
>
> [1] Amro Abbas, Evgenia Rusak, Kushal Tirumala, Wieland Brendel, Kamalika Chaudhuri, and Ari S Morcos. Effective pruning of web-scale datasets based on complexity of concept clusters. arXiv preprint arXiv:2401.04578, 2024.
>
> [2] Noveen Sachdeva, Benjamin Coleman, Wang-Cheng Kang, Jianmo Ni, Lichan Hong, Ed H Chi, James Caverlee, Julian McAuley, and Derek Zhiyuan Cheng. How to train data-efficient llms. arXiv preprint arXiv:2402.09668, 2024.
>
>
> [3] Ben Sorscher, Robert Geirhos, Shashank Shekhar, Surya Ganguli, and Ari Morcos. Beyond neural scaling laws: beating power law scaling via data pruning. Advances in Neural Information Processing Systems, 35: 19523–19536, 2022.
>
>
> **Q3: The norm that is used to measure distance in embedding space is not explicitly stated. Moreover, LLM embedding spaces tend to be highly anisotropic, yet this metric does not seem to adjust for that at all.**
> **A3:** Thank you for your comments.
>
> In this context, we use cosine similarity as the norm. As you mentioned, the space exhibits anisotropy, which is why we incorporate the weight \(\rho_i\) in our density measure (definition (1)). We agree that a single-dimensional metric cannot describe the entire dataset comprehensively. Our work aims to identify a metric that can guide and enhance training efficiency, where the lower density datasets have a smaller performance decay

---

> ### Author Response · Authors · 2024-11-27
> **Response to Reviewer g7pr- part 2**
>
> **Q4: They decay factor in 3.1 is just another hyperparameter to be tuned; it is not all that surprising that adding another degree of freedom allows the curve to better fit the observed pattern. If the decay factor could be calculated automatically, as a function of the data itself, the result would be much more impressive. In fact, the geometric notion of data density not finding its way into the proposed sub-scaling laws suggests that they are not all that useful as a formalism and are yet another way of some data is lower-quality than other data.**
>
> **A4:** Thank you for your suggestions.
>
> Modeling density with decay factor is an interesting idea. However, unlike tokens and model size, which can be designed for different sizes, density is hard to design. We could find two similar samples, but finding two similar datasets is challenging due to the limited combinations, leading to fewer datasets with different densities. This makes modeling the relationship between density and decay factor difficult.We have added experiments on different density datasets, which show that higher density datasets have a larger performance decay, meaning the rate of performance improvement slows down more as the number of tokens increases. Conversely, lower density datasets have a smaller performance decay, meaning the rate of performance improvement slows down less as the number of tokens increases.We will investigate incorporating data density into the proposed sub-scaling laws, where data density will have a relation with the decay factor.
>
>
> **We gratefully appreciate your time in reviewing our paper and your comments. We have made extensive efforts to address your comments and believe that they adequately address all your comments. The reviewer's comments are mainly about some clarifications and are not fatal to the contributions of our manuscript; we believe that the reviewer's insightful comments can be easily and effectively addressed in the final version. We would be grateful if the reviewer could increase the score.**

---

> > ### Comment · Reviewer_g7pr · 2024-11-28
> > **response**
> >
> > Thank you for your replies. I think the core idea of the paper has a lot of promise -- and when I read the abstract I was very excited -- but the current execution just falls short for me. Specifically, the lack of an explicit connection between data density and the final scaling law -- which I understand is not trivial to derive, but is what makes this direction so interesting. Based purely on the high-level idea, I will have to keep my score at 5, but I will defer to the AC if they think the idea on its own has enough promise to overcome its current implementation.

---

### Official Review · Reviewer_JtHD · 2024-11-03

**Soundness:** 1
**Presentation:** 1
**Contribution:** 1
**Rating:** 1
**Confidence:** 3

**Summary:**

The paper proposes a metric to measure “dataset density” (i.e., diversity) in the context of LLM pre-training. It then introduces a scaling law formulation that incorporates such “dataset density” metric. Lastly, the paper introduces another scaling law formulation which accounts for the amount of “overtraining” beyond the number of compute optimal training tokens. The paper includes some experiments aiming to validate the proposed scaling laws.

**Strengths:**

Better understanding the extent to which traditional scaling laws hold at ever larger compute regimes is an interesting and valuable research direction; and so are novel scaling law formulations which better account for current pre-training practices such as 1) performing multiple epochs on the pre-training data (i.e., data repetition) and 2) “over training” beyond Chinchilla compute-optimal.

**Weaknesses:**

The paper has severe limitations. I do not think that it makes substantial contributions with respect to prior work. The writing is at times not very clear. Much of the Appendix appears to be LLM generated. Let me first address each of the “key findings” outlined at the end of the introduction.

> “Sub-Scaling Law Phenomenon: Traditional scaling laws fail to predict performance improvements for very large models and dataset”

Prior work highlights that loss is remarkably predictable, see for instance GPT-4 technical report: “the fitted scaling law predicted GPT-4’s final loss with high accuracy (Figure 1)” or Llama 3 405B technical report “We find this two-step scaling law prediction, which extrapolates over four orders of magnitude, to be quite accurate: it only slightly underestimates the final performance of the flagship Llama 3 model”. Note that performance is even underestimated! This paper does not train models to nearly the scale of GPT-4 or Llama 3 405B, nor does it otherwise present convincing evidence for the purported “sub-scaling” phenomena occurring for “very large models and dataset”.

Unlike loss, benchmark performance can be notoriously hard to predict (e.g., due to emergence). However, “traditional scaling laws” pertain to loss and not benchmark performance. I therefore find Figure 2 (a) highly misleading — no practitioner would expect MMLU accuracy against compute to admit a log-linear fit. This is not a failure of pushing scaling laws to “very large models and datasets”. Log-linear fits of benchmark performance against compute are generally simply not adequate, as is widely known in the literature, see for instance Wei et al. [1].

> “Impact of Data Density: High data density causes sub-scaling due to diminishing returns from redundant data.”

To substantiate their claims, the authors train two models on two datasets, one with higher “data density” than the other. The one with higher “density” happens to perform better. N=2 is clearly insufficient to convincingly study the relationship between “data density” and performance.

Note that the effect of data repetition has been comprehensively studied by Muennighoff et al. [2]. I do not see what this work adds to that of Muennighoff et al.
> Low-density datasets, with more diverse data, align more closely with traditional scaling laws

I believe that the authors aim to substantiate this claim with Figure 8. The experimental set-up here is fundamentally not adequate. Scaling laws do not aim to predict performance throughout training, as is presented in Figure 8. Rather, they aim to predict model performance across different compute scales. Thus, Figure 8 is not informative of the merits of neither “traditional scaling laws” nor the proposed law. Moreover, using intermediate checkpoints for fitting (let alone validating) scaling laws can have severe pitfalls [3, 4].

> “3. Optimal Resource Allocation: Efficient computational resource allocation is crucial to mitigate sub-scaling and sustain performance improvements”

I fail to see how this is a novel insight. Efficient resource allocation is one of the main motivations behind the seminal work of Kaplan et al. [5], which say “For optimal performance all three factors must be scaled up in tandem”. Highly influential subsequent work has discussed better practices for determining such efficient allocation via scaling laws [6]. If the allocation is not optimal, then performance will be worse that it could have been for a given compute scale and no longer follow the scaling law (what in this work is called “sub-scaling”). Figure 9 is highly misleading, since as per Kaplan et al. “Empirical performance has a power-law relationship with each individual factor when not bottlenecked by the other two” — in Figure 9, the larger models are being bottle-necked by not having enough training tokens. Please see Figure 2 of Kaplan et al., such “bottle necks” are not “failures” of “traditional” scaling laws but rather very much to be expected.

> "4. Sub-Optimal Scaling Law: We proposed a Sub-Optimal Scaling Law that generalizes the Chinchilla scaling law to better predict performance and loss in sub-scaling regimes”

Here I do find interesting the inclusion of the “over-training factor” OTR on the scaling law. Note that an identical factor is considered by the concurrent work of Gadre et al. [7]. However, I find the experiments flawed. The limitations of Figure 9 I have already discussed. The results in Figure 10 are also not informative of the merits of the proposed scaling law, since there is a critical flaw: the law is fitted and tested on different pre-training distributions (one being presumably the Pile) and the other being whatever Llama 3 was trained on. Such distribution mismatch thus confounds the fitting, e.g., to what extent is the loss Llama 3 8B over estimated due to such distribution mismatch?

[1] Wei, J., Tay, Y., Bommasani, R., Raffel, C., Zoph, B., Borgeaud, S., ... & Fedus, W. (2022). Emergent abilities of large language models. arXiv preprint arXiv:2206.07682.

[2] Muennighoff, N., Rush, A., Barak, B., Le Scao, T., Tazi, N., Piktus, A., ... & Raffel, C. A. (2023). Scaling data-constrained language models. Advances in Neural Information Processing Systems, 36.

[3] Hoffmann, J., Borgeaud, S., Mensch, A., Buchatskaya, E., Cai, T., Rutherford, E., ... & Sifre, L. (2022). Training compute-optimal large language models. arXiv preprint arXiv:2203.15556.

[4] Porian, T., Wortsman, M., Jitsev, J., Schmidt, L., & Carmon, Y. (2024). Resolving discrepancies in compute-optimal scaling of language models. NeurIPS 2024.

[5] Kaplan, J., McCandlish, S., Henighan, T., Brown, T. B., Chess, B., Child, R., ... & Amodei, D. (2020). Scaling laws for neural language models. arXiv preprint arXiv:2001.08361.

[6] Hoffmann, J., Borgeaud, S., Mensch, A., Buchatskaya, E., Cai, T., Rutherford, E., ... & Sifre, L. (2022). Training compute-optimal large language models. arXiv preprint arXiv:2203.15556.

[7] Gadre, S. Y., Smyrnis, G., Shankar, V., Gururangan, S., Wortsman, M., Shao, R., ... & Schmidt, L. (2024). Language models scale reliably with over-training and on downstream tasks. arXiv preprint arXiv:2403.08540.

**Questions:**

See Limitations.

---

> ### Author Response · Authors · 2024-11-27
> **Response to Reviewer g7pr - part 1**
>
> Dear reviewer JtHD, we appreciate your efforts and detailed feedback very much! However, we believe that there are some misunderstandings. Therefore, we would like to provide a point-by-point response to your comments.
>
>
> **Q1: Sub-Scaling Law Phenomenon: Traditional scaling laws fail to predict performance improvements for very large models and datasets**
>
> **A1**:  We agree that loss is remarkably predictable, however, as pointed out by recent works [1] **there are discrepancies in the traditional scaling laws for predicting loss, which were developed by Kaplan et al. and Hoffmann et al.**  These laws yield substantially different predictions for the optimal model size.
> Moreover, **the equations of scaling law for predicting loss are different in GPT-4 technical report or Llama 3 405B** technical report, these different functions show that the traditional scaling law could not fit all scenarios, the reason is shown in Figure 1 and 9 in our paper: the loss curve could not be estimated using a simple power function.
>
> **Moreover, many recent works [2,3,4,5] provide many different methods to predict the benchmark performance, especially on emergence ability.**
> For example, [2]  show the surprising predictability of complex scaling phenomena using their proposed method:  several emergent phenomena follow a smooth, sigmoidal behavior and are predictable from small models; and show that the agent performance of models such as GPT-4 can be precisely predicted from simpler non-agentic benchmarks. [3] figures out **a clear relationship between the size of the training data and the performance scores across the CEval, CMMLU, and MMLU benchmarks, and could predict performance in CEval, CMMLU, and MMLU benchmarks using their proposed method. [4] present an empirical equation named "Performance Law" to directly predict the MMLU score of an LLM, which is a widely used metric to indicate the general capability of LLMs in real-world conversations and applications. Based on only a few key hyperparameters of the LLM architecture and the size of training data, they obtain a quite accurate MMLU prediction of various LLMs with diverse sizes and architectures developed by different organizations in different years. Performance law can be used to guide the choice of LLM architecture and the effective allocation of computational resources without extensive experiments.**
>
> In addition, our work aims to highlight scenarios where performance improvements decelerate due to factors like data redundancy and non-optimal resource allocation, which may not be fully captured by traditional scaling laws.
>
> We will ensure our claims are more clearly positioned within the context of existing findings and provide additional evidence to support the sub-scaling phenomenon.
>
>
> [1] Porian T, Wortsman M, Jitsev J, et al. Resolving discrepancies in compute-optimal scaling of language models[J]. arXiv preprint arXiv:2406.19146, 2024.
>
>
> [2] Ruan Y, Maddison C J, Hashimoto T. Observational Scaling Laws and the Predictability of Language Model Performance[J]. arXiv preprint arXiv:2405.10938, 2024.
>
> [3] Yang C, Li J, Niu X, et al. The Fine Line: Navigating Large Language Model Pretraining with Down-streaming Capability Analysis[J]. arXiv preprint arXiv:2404.01204, 2024.
>
> [4] Chuhan Wu and Ruiming Tang. Performance law of large language models. arXiv preprint arXiv:2408.09895,
> 2024.
>
> [5] Berivan Isik, Natalia Ponomareva, Hussein Hazimeh, Dimitris Paparas, Sergei Vassilvitskii, and Sanmi Koyejo.
> Scaling laws for downstream task performance of large language models. arXiv preprint arXiv:2402.04177,
> 2024.
>
> Q2: **Impact of Data Density: High data density causes sub-scaling due to diminishing returns from redundant data**
>
> A2: Thanks for your suggestion!
>
> Here, we want to emphasize the **distinction between density and repetition. Density does not refer to multiple epochs but rather to different data distribution scenarios. For example, in Figure 5, the three sentences within the blue circle have similar contents. In Muennighoff et al. [4], such sentences would not be considered repetitive.** However, our study investigates whether these content-repetitive but non-identical sentences, under the same amount of training tokens, affect training efficiency. This is the fundamental difference from Muennighoff et al. [6]. In essence, with the same quantity of data, higher content repetition leads to increased density, which directly results in reduced diversity, thereby impacting the model's training efficiency.
>
> We agree that the current experiments with N=2 are insufficient to conclusively study the relationship between data density and performance. We will expand our experiments to include a larger number of datasets with varying densities to provide more robust evidence.
>
> [6] Muennighoff (2023). Scaling data-constrained language models. Advances in Neural Information Processing Systems, 36.

---

> ### Author Response · Authors · 2024-11-27
> **Response to Reviewer g7pr - part 2**
>
> **Q3: Low-density datasets, with more diverse data, align more closely with traditional scaling laws**
>
> **A3:** Thank you for your insightful comments. We appreciate the opportunity to clarify our approach and the intent behind Figure 8.
>
> - The primary goal of Figure 8 was to illustrate the relationship between different density datasets with training data size and model performance over various training checkpoints, rather than to validate the traditional scaling laws directly. We aimed to show how model performance evolves with increased training data and how closely it aligns with our proposed sub-scaling law's predictions at different stages of training.
> - Traditional scaling laws, as you correctly pointed out, are designed to predict model performance across different compute scales, typically focusing on final performance metrics after extensive training. Our analysis extends this concept to observe performance trends at intermediate stages, providing a more granular view of the training dynamics.
> - While traditional scaling laws offer valuable insights into the final performance, understanding the intermediate checkpoints can help in optimizing training processes, identifying early signs of convergence or divergence, and making informed decisions about resource allocation during training.
> - Our approach aims to bridge the gap between traditional scaling laws and practical training observations. While traditional scaling laws focus on asymptotic performance, our analysis provides actionable insights during the training process.
>
> **Q4: Optimal Resource Allocation: Efficient computational resource allocation is crucial to mitigate sub-scaling and sustain performance improvements**
>
> **A4:**
> Thanks for your suggestion.
> We recognize that the importance of efficient resource allocation is well-established in the literature, particularly in the work of Kaplan et al. [7].
> In our paper, we focus on how different training strategies influence the sub-scaling law, training strategies includes model/data allocation and the selection of hyper-parameters such as batch size and learning rate. .
> Hyper-parameters are critical factors controlling the learning process of the model, with batch size and learning rate being the most important. However, their selection does not significantly impact sub-scaling phenomena. Non-optimal hyper-parameters tend to show poor performance from the beginning of the training process rather than causing a deceleration in performance improvement with the dataset or model size increases, and thus do not lead to the sub-scaling effect,
>
> We will ensure our discussion acknowledges prior work more explicitly and clarifies our contribution in the revised paper.
>
>
>
>
>
>
> **Q5: Sub-Optimal Scaling Law: We proposed a Sub-Optimal Scaling Law that generalizes the Chinchilla scaling law to better predict performance and loss in sub-scaling regimes**
>
> **A5:**
>
> We appreciate the interest in our inclusion of the Over-Training Ratio (OTR). To address concerns about the distribution mismatch between pre-training datasets, we test the performance of Llama 3 on the Pile dataset. As highlighted in previous work by Kaplan et al. (2020), "When we evaluate models on text with a different distribution than they were trained on, the results are strongly correlated to those on the training validation set with a roughly constant offset in the loss – in other words, transfer to a different distribution incurs a constant penalty but otherwise improves roughly in line with performance on the training set."
>
> Therefore, we believe that such a distribution shift does not significantly influence our predictions. The constant offset in loss due to the distribution shift is accounted for, ensuring that the overall trend and performance improvements remain valid and comparable. This supports the robustness of our proposed sub-optimal scaling law even when applied across different data distributions.
>
> [7] Kaplan J, McCandlish S, Henighan T, et al. Scaling laws for neural language models[J]. arXiv preprint arXiv:2001.08361, 2020.

---

### Official Review · Reviewer_gSiV · 2024-11-04

**Soundness:** 3
**Presentation:** 2
**Contribution:** 3
**Rating:** 6
**Confidence:** 3

**Summary:**

This paper explores the phenomenon of sub-scaling laws by investigating a total of 400 models, ranging from 2 million to 7 billion in parameter size, motivated by observations that model performance gains decelerate for increasing model sizes. The authors focus on two perspectives for this, training data density (defined through measures reflecting data similarity and redundancy in training corpora) and training strategy (defined through optimality of resource allocation during training).

**Strengths:**

The paper provides insightful findings around training dynamics of large language models encompassing various sizes. The analysis is overall extensive, and the results contribute to a better understanding of experimental design decisions for the training of LLMs. As such, I believe this paper represents a solid contribution that is of interest to the research community.

**Weaknesses:**

* I found the paper overall difficult to read, the structure leaves room for improvement and the findings / conclusions could be better depicted and illustrated. The paper focuses heavily on technical derivations and definitions and leaves several experimental details unmentioned in the main manuscript (I formulate these as questions below).
* I would have wished the authors to better explain the implications of the findings provided in this work. How can future research contribute to better understanding the phenomenon of sub-scaling laws?
* Re. presentation style: Figure 1 could be more clearly explained (e.g., the legend is incomplete), Figure 5 is confusing (the blue and yellow clusters look very similar both w.r.t. their size and the number / distribution of points within them, even though they represent a cluster with high and low similarity / density, respectively), and if citations aren’t used actively, I suggest surrounding them with parentheses for better readability.

**Questions:**

* Can you provide additional details on Figure 6? How was the data clustered, how was the number of clusters chosen?
* Where does the constant for the definition of $\text{FLOPs}(N, D)$ come from? Can you elaborate on its derivation?
* For OTRs, how is $D_{opt}$ derived? This is unclear in the paper but quite crucial to understand the authors’ argumentation around the training strategy.

---

> ### Author Response · Authors · 2024-11-28
> **Response to Reviewer gSiV**
>
> Thank you for the valuable feedback. We appreciate the opportunity to address the concerns raised and clarify the details and methodologies of our paper.
>
> **Q1: Readability and Structure**
>
> **A1:** We acknowledge that the paper's readability and structure can be improved. We will revise the manuscript to ensure a clearer structure, including better segmentation of sections and a more logical flow of ideas. We will also enhance the depiction and illustration of findings and conclusions, ensuring that they are easily understandable. Additionally, we will include more experimental details in the main manuscript to provide a comprehensive understanding of our work.
>
> **Q2: Implications of Findings**
>
> **A2:** We appreciate your suggestion to better explain the implications of our findings. The phenomenon of sub-scaling laws highlights the limitations of traditional scaling laws when applied to very large models and datasets.
> Future research can contribute to a better understanding of sub-scaling laws by:
> - Investigating the impact of different data densities and training strategies on model performance.
> - Developing new metrics and methodologies to measure and predict sub-scaling effects.
> - Exploring the role of optimal resource allocation in mitigating sub-scaling and enhancing model efficiency.
> - Conducting large-scale experiments to validate and refine the proposed sub-scaling laws across diverse datasets and model architectures.
>
> **Q3: Presentation Style**
>
> **A3:** Thanks for your suggestion. We will revise the presentation of figures to improve clarity:
> - **Figure 1**: We will provide a more detailed explanation and complete the legend to ensure all elements are clearly described.
> - **Figure 5**: We will refine the visual representation to distinctly differentiate between high and low similarity/density clusters. We will use different colors or patterns to make the differences more apparent.
> - **Citations**: We will use parentheses to surround citations for better readability and ensure they are actively integrated into the text.
>
> **Q4: Additional Details on Figures**
>
> **A4:** We will provide additional details on the clustering process used in Figures: Figures 5 and 6 are motivated by Abbas et al. We followed their study by using Faiss as the clustering method. The choice of the number of clusters is based on empirical experience. We used the sqrt to roughly determine the number of clusters, which is 14374. Cluster IDs are discontinuous. We will clarify it in our paper.
>
>
> **Q5: Constant for FLOPs(N,D)**
>
> **A5:** The constant for the definition of FLOPs(N,D) is derived from empirical observations and theoretical considerations of the computational complexity involved in training large language models. Specifically, the formula FLOPs(N,D) ≈ 6ND is based on the relationship between the number of parameters (N) and the number of training tokens (D), reflecting the compute budget required for training. We will provide a detailed derivation and reference the relevant literature to support this constant.
>
> **Q6: Derivation of D_opt for OTRs**
>
> **A6:** The optimal number of training tokens (D_opt) is derived based on the compute-optimal training strategy proposed by Hoffmann et al. (2022). D_opt is determined by balancing the model size (N) and the number of training tokens (D) to achieve the best performance without over-training.
> We will provide a clearer explanation of how D_opt is calculated, including the theoretical basis and empirical validation, to ensure a comprehensive understanding of its role in our training strategy.

---

> > ### Comment · Reviewer_gSiV · 2024-12-02
> >
> > Many thanks to the authors for the detailed response. I appreciate the clarifications and will keep my positive score.

---

### Official Review · Reviewer_a2Th · 2024-11-04

**Soundness:** 3
**Presentation:** 3
**Contribution:** 3
**Rating:** 6
**Confidence:** 3

**Summary:**

These work systematically investigate the sub-scaling law under over-training scenarios, and propose the new formulations with OverTraining Ratio OTR. The observations on data density and data/model size allocations are interesting and convincing. The fit curves seem better than the current widely-used ones, especially under over-training conditions for SLMs.

**Strengths:**

1. This paper is well-organized and easy to follow.

2. It is an important and interesting topic to explore sub-scaling phenomenon.

3. Extensive experiments and careful analyses demonstrates the effectiveness of the proposed new formulations.

**Weaknesses:**

1. Some details are missing. For example, in both abstract and introduction, the paper mentions over 400 models, ranging from 20M to 7B, but we can find no details of the used model. Also, many citations are missing. Some claims (e.g., line 280， 763） and existing formulations （e.g., line 307, ) should add citations to support the claims and separate from the proposed formulations.

2. When a notation first comes up, there should be clearly description, or at least a clear reference to later sections. For example, R_D and R_N are two very important notations. When I read line 369, I thought R_D might be a hyper-parameter. Then when I read line 456, R_D and R_N seem to be two functions, but the exact definition is just mentioned as in Appendix in line 456 without the clear reference. Finally, I found it in Appendix D.  Additionally, for “optimal training resource allocations”, I fully understood this concept until section 2.2.

3. The description should be consistent through the paper. For example, in the relative works, it is said that the difference to current works are they focus on general scaling laws while this work provides under over-training conditions (line 1060-1061). However, main sections seem not convey this message.

**Questions:**

1. What are these 400 models?

2. What’s the relationship of Eq. (6) and (7)?

3. In the appendix D, the final results actually only related to OTR. Why the OTR can both represent data density and data/model size allocations?

---

> ### Author Response · Authors · 2024-11-28
> **Response to Reviewer a2Th**
>
> Thank you for the valuable feedback. We appreciate the opportunity to address the concerns raised and clarify the details and methodologies of our paper.
>
> **Q1: Missing Details of 400 Models and Citations**
>
> **A1:** We apologize for the lack of details and citations in the abstract and introduction. We will revise the paper to include comprehensive details about the 400 models used in our experiments, ranging from 20 million to 7 billion parameters. The architecture details of the used models are listed in Appendix Table 3. We run them with varying datasets and training strategies. We train 400 models in different density datasets, and with different model/data allocation and the selection of hyper-parameters, to investigate the sub-scaling law phenomenon.  Some claims (e.g., line 280， 763）about optimal batch size have been revised according to previous work for clarity [1, 2], and we have added the citation to be more accurate.
>
> Additionally, we will ensure that all claims and existing formulations are properly cited to support our arguments and clearly distinguish between proposed formulations and those from prior work. For example, the formula C = 6ND(Line 307), it is used for an estimate of the total non-embedding training computation and been used in previous work [1] will be supplemented with appropriate references.
>
> **Q2: Clear Descriptions of Notations**
>
> **A2:** Thanks for your valuable advice on the manuscript structure! We have taken your advice and moved the detailed definition of R_N and R_D from the appendix to the manuscript. Similarly, we will clarify the concept of "optimal training resource allocations" earlier in the paper to aid reader understanding, which has been described in previous works [1, 3]. The amount of training tokens is D_{opt}. More description about this problem has been replenished in our manuscript.
>
>
> **Q3: Consistency in Description**
>
> **A3:** Thanks for your careful reviews. The discrepancy between descriptions in main sections and related work has been checked carefully. We have revised the inconsistent words like 'over-training condition' and replaced them with 'sub-optimal scaling condiction'.
>
> **Q4: Relationship Between Eq. (6) and (7)**
>
> **A4:** Equation (6) models the performance \(P(n)\) as a function of the number of samples \(n\), incorporating the information gain \(I(n)\). Equation (7) further refines this model by introducing decay factors \(R_D\) and \(R_N\) to account for the influence of data density and model size on performance. We will clarify the relationship between these equations, explaining how Eq. (7) generalizes Eq. (6) by incorporating additional factors to better predict performance in sub-scaling regimes.
>
> **Q5: OTR and data density**
>
> **A5:** In Appendix D, the final results are related to the Over-Training Ratio (OTR). The OTR can represent both data density and data/model size allocations because it encapsulates the relationship between the amount of training data and the model size. High OTR values indicate over-training, where the amount of data exceeds the optimal allocation for a given model size, leading to performance degradation.
> In our paper, we study how training strategies, including hyper-parameters and data/model allocation (OTR), as well as data density, influence the sub-scaling law.
> We will provide a more detailed explanation of how OTR captures these aspects and its role in our proposed sub-scaling law.
>
>
> [1] Kaplan J, McCandlish S, Henighan T, et al. Scaling laws for neural language models[J]. arXiv preprint arXiv:2001.08361, 2020.
>
> [2] Hu S, Tu Y, Han X, et al. Minicpm: Unveiling the potential of small language models with scalable training strategies[J]. arXiv preprint arXiv:2404.06395, 2024.
>
> [3] Hoffmann J, Borgeaud S, Mensch A, et al. Training compute-optimal large language models[J]. arXiv preprint arXiv:2203.15556, 2022.

---

### Note · Authors · 2024-12-18

I have read and agree with the venue's withdrawal policy on behalf of myself and my co-authors.